# Evaluating the Use of Neonatal Colonization Screening for Empiric Antibiotic Therapy of Sepsis and Pneumonia

**DOI:** 10.3390/antibiotics12020189

**Published:** 2023-01-17

**Authors:** Alisa Bär, Sabina Schmitt-Grohé, Jürgen Held, Julia Lubig, Gregor Hanslik, Fabian B. Fahlbusch, Heiko Reutter, Joachim Woelfle, Adriana van der Donk, Maria Schleier, Tobias Hepp, Patrick Morhart

**Affiliations:** 1Division of Neonatology and Pediatric Intensive Care, Department of Pediatrics and Adolescent Medicine, Friedrich-Alexander-University of Erlangen-Nuernberg, Loschgestraße 15, 91054 Erlangen, Germany; 2Division of Allergology, Pulmonology and Mucoviscidosis, Department of Pediatrics and Adolescent Medicine, Friedrich-Alexander University of Erlangen-Nuernberg, Loschgestraße 15, 91054 Erlangen, Germany; 3Institute of Clinical Microbiology, Immunology and Hygiene, University Clinic Erlangen, Wasserturmstraße 3/5, 91054 Erlangen, Germany; 4Institute of Human Genetics University Clinic Erlangen, Friedrich-Alexander University of Erlangen-Nuernberg, 91054 Erlangen, Germany; 5Institute for Medical Informatics, Biometry and Epidemiology (IMBE), Friedrich-Alexander-University Erlangen-Nuernberg, Waldstr. 6, 91054 Erlangen, Germany

**Keywords:** multidrug-resistant Gram-negative bacteria, MDRGN, multidrug-resistant bacteria, MDR, neonates, colonization, neonatal infection, sepsis, pneumonia, screening

## Abstract

(1) Background: Since 2013, weekly screening for multidrug-resistant Gram-negative (MDRGN) bacteria has been performed in German neonatal intensive care units (NICU). National guidelines recommend considering these colonization analyses for antibiotic treatment regimens. Our retrospective single center study provides insight into the clinical dichotomy of bacterial colonization and infection rates in neonates. (2) Methods: We analyzed microbiological data of neonates admitted to our tertiary level NICU over nine years. Colonization with MDRGN/*Serratia marcescens* (SERMA) was compared to microbiological findings in sepsis and pneumonia. (3) Results: We analyzed 917 blood and 1799 tracheal aspirate samples. After applying criteria from the Nosocomial Infection Surveillance for Neonates (NEO-KISS), we included 52 and 55 cases of sepsis and pneumonia, respectively; 19.2% of sepsis patients and 34.5% of pneumonia patients had a prior colonization with MDRGN bacteria or SERMA. In these patients, sepsis was not attributable to MDRGN bacteria yet one SERMA, while in pneumonias, ten MDRGN bacteria and one SERMA were identified. We identified late-onset pneumonia and cesarean section as risk factors for MDRGN/SERMA acquisition. (4) Conclusions: Colonization screening is a useful tool for hygiene surveillance. However, our data suggest that consideration of colonization with MDRGN/SERMA might promote extensive use of last resort antibiotics in neonates.

## 1. Introduction

Newborns admitted to the neonatal intensive care unit (NICU) are at high risk of nosocomial infections [1]. Previous studies suggested two major reasons for the increased susceptibility to bacterial infection especially in preterm infants with low birth weight, namely the immaturity of the neonatal immune system [2] and the prolonged bacterial exposure associated with long hospitalization, including repeated execution of invasive procedures [3]. Although development of cell-mediated immunity is taking place in the first trimester of pregnancy, neonatal humoral immunity is reliant on transplacental passage of IgG, which starts at gestational week 32. Subsequently, especially preterm neonates below this gestational age show significantly reduced levels of circulating IgG antibodies [4]. In 2013, the German commission for hospital hygiene and prevention of infection (KRINKO) released a recommendation for the protection of neonates, underscoring the severe vulnerability of these patients for colonization and infection by multidrug-resistant Gram-negative (MDRGN) bacteria. Against the background of increasing nosocomial bacterial infection rates [5,6,7,8], it is recommended to consider the implementation of a colonization screening in terms of an anal and pharyngeal swab for all neonates upon admission and weekly thereafter for inpatients on German NICUs depending on their structure and patient volume. The KRINKO screening follows two objectives: First, the early detection of nosocomial transmissions and the consequential implementation of preventive hygiene policies. Second, guidance of medical decision-making regarding empirical or calculated therapy of infections based on prior bacterial colonization in the context of suspected infection (i.e., antibiotic stewardship) [7,9]. However, the role of active surveillance for multidrug-resistant (MDR) bacterial colonization in antibiotic stewardship (ABS) programs awaits final evaluation, with controversial discussion of the potential correlation between colonization with resistant bacteria and infection caused by the very same germs in the literature [10,11,12]. While on the one hand, validation of such coherence might have major implications for ABS in the treatment of infectious diseases [13,14], a lack of correlation on the other hand might imply overtreatment with last resort antibiotics for all colonized neonates [15,16]. We therefore aimed to clarify this dichotomy using retrospective microbiological and medical data from our tertiary NICU located in the Department of Pediatrics and Adolescent Medicine of the Friedrich-Alexander-University of Erlangen-Nuernberg, Germany. In addition, we investigated possible risk factors for acquiring colonization with MDR bacteria and MDR-caused infections. We collected data from all neonatal patients from the introduction of the screening in 2013 until the end of 2021 to answer the following research questions:

RQ1: Is colonization with MDR bacteria a risk factor for MDR sepsis and MDR pneumonia in neonates?

RQ2: What risk factors lead to higher likelihood of acquiring colonization with MDR bacteria and MDR-caused infections?

## 2. Methods

### 2.1. Definitions

The KRINKO definition of MDRGN distinguishes bacteria by their in vitro sensitivity against four main classes of antibiotics: (1) acylureidopenicillins, (2) third/fourth generation cephalosporins, (3) carbapenems, and (4) fluoroquinolones. Bacteria which show resistance against three out of the four (four out of the four) antibiotic groups are classified as 3MRGN (4MRGN). Both 3- and 4MRGN are considered as MDR in adults. Due to the restricted applicability of Fluoroquinolones in children, KRINKO introduced the category 2MRGN NeoPaed which consists of bacteria that possess resistance against acylureidopenicillins and third/fourth generation cephalosporins [9]. The weekly colonization screening in our clinic covers bacteria with multidrug resistances such as 2MRGN NeoPaed, 3MRGN, and 4MRGN. Additionally, following the KRINKO recommendation, our screening includes *Serratia marcescens* (SERMA) which possesses a high risk for nosocomial spread and outbreaks especially in the neonatological field. [9,16,17]. Since we also test high-risk patients for methicillin-resistant *Staphylococcus aureus* (MRSA) and vancomycin-resistant enterococcus (VRE), we also included these species in our analysis. In the following analysis, we summarized MRSA, VRE, 2MRGN NeoPaed, 3MRGN, and 4MRGN in the category MDR. As infections, we defined sepsis and pneumonia according to the German Neonatal Hospital Infection Surveillance System NEO-KISS criteria (National Reference Center for Nosocomial Infection Surveillance at the Institute for Hygiene and Environmental Medicine Charité—University Medicine Berlin; www.nrz-hygiene.de), as described below.

For a microbiologically confirmed diagnosis of sepsis, a pathogen that is not a coagulase-negative *Staphylococcus* (CoNS) must be detected in the blood or cerebrospinal fluid and two of the following criteria must be met: fever (>38 °C) or temperature instability or hypothermia (<36.5 °C); tachycardia (>200/min) or new/increased bradycardia (<80/min); prolonged recapillarization time (>2 s); new or increased apneas (>20 s); unexplained metabolic acidosis (BE ≤ 10 mval/L); new onset hyperglycemia (>140 mg/dL); other signs of sepsis. If CoNS was the only pathogen isolated from the blood, two of the criteria just mentioned plus one of the following laboratory parameters must be met: CRP > 2.0 mg/dL or IL6-8 pathologically elevated; platelets < 100/nL; I/T ratio > 0.2; leukocytes < 5/nL. Pneumonia was defined as radiologic finding of new or progressive infiltrates, shadowing, or fluid in the interlobar or pleural space plus worsening of gas exchange with a decrease in oxygen saturation and four of the following criteria: new/increased bradycardia (<80/min) or new/increased tachycardia (>200/min); new/increased tachypnea (>60/min) or new/increased apnea (>20 s); purulent tracheal aspirate; germ from tracheal aspirate; new/increased dyspnea (retractions, nostrils, moans); temperature instability/fever/hypothermia; increased respiratory secretions; CRP > 2.0 mg/dL or interleukin 6–8 pathologically elevated; I/T ratio > 0.2 [18,19].

In our study period from 2013 until 2021, we treated 5988 neonates in our neonatology of which 485 had a very low birth weight (VLBW). We started our analysis by scanning over 26,000 microbiologic culture results from these patients. Whenever a blood culture analysis was performed during that period, it was regarded as a marker for suspected sepsis. Subsequently, the above sepsis criteria were applied to all patients with positive blood cultures (BC). Accordingly, we approached tracheal aspirate (TA) in patients meeting the pneumonia criteria (illustrated in Figure 1 and Figure 2).

### 2.2. Patients and Setting

In this retrospective analysis, we included data from all neonatal patients treated in our division from 1 January 2013 until 31 December 2021. The following data were recorded for each patient: clinical manifestation of sepsis or pneumonia, date of diagnosis, gestational age, adjusted gestational age, sex, mode of delivery, and results of microbiological examinations. Our clinic is a level 3 perinatal center. The division of neonatology comprises three neonatological wards: A NICU with 14 treatment beds for ventilation, a 16-bed neonatal intermediate care unit for non-ventilated newborns, and a 6-bed neonatal surveillance unit located in vicinity to the delivery room in the obstetrical center of the Department of Gynecology and obstetrics of our University hospital.

### 2.3. Hygienic Management

Following the KRINKO guideline of 2013, our clinic decided to introduce a colonization screening which includes every neonatal intensive care patient on all of our three neonatal wards. We test all neonates with an estimated stay of at least 12 h upon admission and then weekly by anal and pharyngeal swabs irrespective of their gestational age and birth weight. In addition, all neonates are tested on inpatient admission. In suspicion of an infection or nosocomial transmissions, shortened screening intervals apply. Additionally, a bacteriological screening is taken from tracheal aspirate from all neonatal patients on ventilation after intubation or admission and continued twice weekly. A parents’ questionnaire supports the identification of high-risk patients for MRSA colonization. Whenever there is a positive medical history indicating a high prevalence background such as contact to farm animals, necessity of dialysis, chronic skin lesions, or hospitalization in the past 12 months, we perform a pharyngeal swab test for MRSA. VRE screening is solely performed for neonates with VRE history and for all neonates in case of an infection on the ward. A colour coded alert system assures sufficient hygienic management on the ward: At the time of admission, neonates remain isolated by bed location and are treated as potentially colonized. After receipt of the microbiological screening results (up to 48 h), patients are either isolated in a single room (red category) in case of 3- or 4MRGN or MRSA, remain isolated by bedside in case of 2MRGN NeoPaed (yellow category) or isolation is set aside (green category, negative test results).

### 2.4. Microbiological Diagnostics

After collection, blood cultures (BC), tracheal aspirates (TA), and pharyngeal and anal swabs were stored at room temperature and transported to the Microbiology Institute six times a day. Specimens taken at night were stored at 4 °C and blood cultures were stored at room temperature until transport the next day.

For MRGN screening, rectal and pharyngeal samples were placed on C3GR agar (Mast Diagnostica GmbH, Reinfeld, Germany) with a ceftriaxone disk and incubated at 37 °C for a minimum of 24 h. For detection of SERMA, samples were applied to an endo agar with a cefazoline disk and incubated at 37 °C for a minimum of 24 h. Blood and tracheal samples underwent standard bacteriological examination, i.e., blood samples were streaked out on sheep blood agar, chocolate agar, and tryptic soy broth (TSB) followed by incubation for five days at 37 °C with 5% CO_2_. TAs were incubated on sheep blood agar, chocolate agar, and endo agar for at least 48 h at 37 °C with 5% CO_2_ and tested for all potential pneumonia pathogens. Identification of isolates was performed using the Bruker MALDI Biotyper^®^ (MBT) smart (Bruker Daltonik GmbH, Bremen, Germany). Antimicrobial susceptibility testing was carried out with a VITEK^®^ 2 (bioMérieux GmbH, Bruchsal, Germany).

### 2.5. Statistical Analysis

The statistical analysis was carried out by the Institute for Medical Informatics, Biometry and Epidemiology (IMBE) of Friedrich-Alexander-University Erlangen-Nürnberg, Germany. The correlation between MDR bacteria and SERMA in anal/pharyngeal swabs and BCs/TAs was determined by using McNemar Test for paired samples corrected by Chi^2^ continuity correction. We calculated the sensitivity, specificity, positive predictive value (PPV), and negative predictive value (NPV) for MDR bacteria and SERMA for all rectal and pharyngeal swabs and BC/TA pairs in all neonates with sepsis/pneumonia. In case of PPV, for example, we calculated the relation of true positives (MDR bacteria/SERMA isolate found in screening and BC/TA) amongst all MDR bacteria/SERMA isolates in anal and pharyngeal screening. We discriminated between late- and early-onset sepsis/pneumonia (LOS/EOS, LOP/EOP), gestational age at birth, adjusted gestational age, delivery mode, and sex to identify other variables in correlation with MDR bacteria/SERMA in screening and as infectious agents in terms of infection. For significance test we calculated Chi^2^, or Fisher’s exact test depending on the expected frequencies. We assessed the contingency coefficient, odds ratio, and 95%-confidence interval to describe the strength of correlation. All *p*-values < 0.05 were considered statistically significant.

## 3. Results

### 3.1. Study Selection and Description

The analysis of BCs and sepsis included 917 blood samples of 546 patients (Figure 1). These contained 136 samples (14.8%) with positive results from 117 (21.4%) independent individuals. We excluded specimens of patients that missed a screening (29) or were not from neonates/premature infants (older than 28 days at admission) (9). A total of 59 samples (6.4%) of 52 (9.5%) neonates met the NEO-KISS criteria of sepsis. Seven cases were further excluded because of multiple positive BCs, leading to 52 (5.7%) blood samples of 51 (9.3%) neonates with clinically manifested sepsis. Our study selection for the sepsis study arm is illustrated in Figure 1.

The analysis of TAs and pneumonia included 1799 samples of TA from 610 neonates (Figure 2). These contained 744 (41.4%) with positive results from 318 (52.1%) independent individuals. Analogously to the sepsis arm of our study the following patients were omitted: neonates without performed screening (135), patients older than 28 days of life (87), or with missing data (1). Of the remaining 521 samples, 375 did not fulfil NEO-KISS criteria of pneumonia. We rejected multiple TAs per neonates in line with the treatment information leading to 55 (3.1%) samples of positive TA from 54 (8.9%) neonates. A flowchart of the study selection for the pneumonia study arm is illustrated in Figure 2.

### 3.2. Correlation between Colonizing and Infecting Bacteria

Of the 52 samples with positive BC and sepsis, 19.2% (*n* = 10) showed positive microbiological screening results (Figure 3). Of those, 60% (*n* = 6) were MDR bacteria, while 40% (*n* = 4) showed SERMA. Of the 6 MDR bacteria positive screening samples, 83.3% (*n* = 5) were solely found in anal swab and in only one case, there were MDR bacteria also in the pharyngeal swab. All SERMA positive neonates showed both anal and pharyngeal colonization. There was one case of 3MRGN, but no cases of MRSA and VRE in the colonization surveillance. There was not a single case of MDR bacteria positive BC, but one SERMA positive BC. Based on the available numbers, we executed the McNemar Test corrected by continuity correction (*p* = 0.008). This significant result suggests that bacterial screening has not been useful in predicting sepsis caused by MDR bacteria or SERMA. We calculated the overall sensitivity, specificity, PPV, and NPV of detecting bacteria with special resistances and multidrug resistances in rectal and pharyngeal swab samples compared to those in BC samples. The results were 100% for the overall sensitivity, 82.3% for the specificity, 10% for PPV, and 100% for NPV.

Of the 55 samples of positive TA and pneumonia, 34.5% (*n* = 19) showed at least one relevant bacterium in colonization screening (Figure 4). In 57.9% (*n* = 11) of the colonized neonates, we found resistant bacteria in TA. We detected one case of SERMA in TA. We observed 14 cases of 2MRGN NeoPaed, two cases of 3MRGN, one case of MRSA, two cases of SERMA, and no VRE in the colonization surveillance. We isolated six different species of 2MRGN NeoPaed and two of 3 MRGN. Of the 11 cases where MDR bacteria or SERMA were found both in the colonization screening and TA concomitantly, there was only one case with exclusively positive anal swab, two cases with solely positive pharyngeal swab, and eight cases with positive findings in both swabs.

The McNemar test corrected by continuity correction (*p* = 0.013) showed a significant result. In line with the findings regarding sepsis (see above), these results suggest that colonization screening was not suited to predict pneumonia by MDR bacteria or SERMA. We calculated an overall sensitivity, specificity, PPV, and NPV of detecting MDR bacteria and SERMA in rectal and pharyngeal swab samples compared to TA samples. The results were of 100% for the overall sensitivity, 81.8% for the specificity, 57.8% for PPV, and 100% for NPV.

### 3.3. Infectious Agents of Sepsis and Pneumonia

Because two different bacteria were identified concomitantly in one BC, we detected 53 different bacterial species in our 52 samples of BCs in total. On this account, percentages given do not add up to 100 percent. In 52 cases of blood culture proven sepsis, we detected in 65.4% (*n* = 34) of the samples coagulase-negative *Staphylococci* (CoNS), in 7.7% (*n* = 4) *Enterococcus* spp., in 5.8% (*n* = 3) *Escherichia coli*, *Staphylococcus aureus,* and *Klebsiella* spp., respectively; in 3.9% (*n* = 2) *Streptococcus agalactiae* and in 1.9% (*n* = 1) *Bacillus cereus*, SERMA, and *Enterobacter cloacae*. We did not detect cases of sepsis with MRGN, VRE, or MRSA. Among our 52 cases of sepsis there was solely one patient with early-onset sepsis (<72 h after birth) caused by group B *Streptococcus*.

Because of multiple species in one sample, we found 77 different bacterial species in our 55 samples of TA. In analogy to our sepsis findings, percentages do not subsequently add up to 100 percent. In 25.5% (*n* = 14) of the patients’ TA, we detected CoNS, in 20.0% (*n* = 11) saprophytic germs, in 16.4% (*n* = 9) 2- or 3MRGN, in 14.6% (*n* = 8) *Enterococcus* spp. and *Klebsiella* spp., respectively; and in 9.1% (*n* = 5) *Escherichia coli*. As well as the 9 cases of 2/3MRGN, there was one case of MRSA and one case of SERMA detected. In total, MDR bacterial- or SERMA-associated pneumonia accounted for 20.0% (*n* = 11) of all pneumonia cases. An overview of the bacteria identified in BC and TA is given in Table 1.

### 3.4. Identification of Risk Factors for Acquiring MDR and SERMA for Patients with Pneumonia

Based on our results so far, we investigated potential risk factors for colonization or infection with MDR bacteria or SERMA. Based on our data in the sepsis study arm, our investigated possible risk factors do not seem to play an important role for acquiring MDR bacteria or SERMA colonization or infection. Hence, we solely listed potential risk factors for neonates with pneumonia in Table 2. We could not detect a significant correlation between gestational age at birth and the acquisition of relevant bacteria (*p* = 0.501). Likewise, the time of infection adjusted gestational age had no significant correlation (*p* = 0.296) either.

In contrast, the type of pneumonia in terms of time of infection showed significant influence (*p* = 0.043). Neonates with late-onset pneumonia (LOP) after the first seven days of life showed an increased susceptibility to MDR bacteria/SERMA compared to patients with early-onset pneumonia (EOP) (OR = 4.80, 95% CI 0.955–24.2). The contingency coefficient (CC = 0.264) displayed a moderate correlation between these two factors. Newborns that were delivered via vaginal birth possessed significantly less colonization with MDR bacteria or SERMA compared to neonates delivered via caesarean section (*p* = 0.027, OR = 0.181, 95% CI 0.0357–0.917). There is a moderate strength of correlation between delivery mode and colonization (CC = 0.293). In contrast, the patients’ sex showed no statistically significant influence (*p* = 0.992).

## 4. Discussion

In this study, we showed that MDR bacteria and SERMA are prevalent nosocomial colonizers in neonates. Among the 52 cases with sepsis, 11.5% (*n* = 6) showed prior colonization with MDR bacteria and 7.7% (*n* = 4) with SERMA. Our analysis (McNemar test) shows that positive neonatal microbiological screening results are ineffective in predicting sepsis and pneumonia caused by MDR bacteria or SERMA.

In neonatal sepsis, SERMA was responsible for only a single patient colonized with the bacteria, while no such relation was observed for 2- or 3MRGN colonization. In opposition, neonatal pneumonia seemed to show a more cohesive bacterial pattern of colonization (swabs) and infection (TA), i.e., of the 34.5% colonized patients (30.9% MDR, 3.6% SERMA), concordant TA isolates were found in 57.9% of the cases. Nevertheless, our significant result in McNemar test (similarly to the sepsis study arm) showed that colonization screening is not adequate in predicting pneumonia with MDR or SERMA, although we detected MDR and SERMA in tracheal aspirate in 20.00% of the pneumonia cases and the PPV (57.9%) was higher than in the sepsis arm.

Previous studies in neonates also indicated that MDR bacterial screenings are limited in predicting infections in terms of sepsis. A study from Halle in Germany from 2014 detected a MDR bacterial colonization rate of 4.9%, of which only 3.2% developed into sepsis [20]. Another study from Germany with comparable colonization rates (i.e., 12.7% of 584 screened neonates) showed similar results to our study with 2 of 23 episodes of sepsis caused by 2MRGN and MRSA, respectively, and a total infection frequency of 1.9% (one case of SERMA) [21]. A study from Taiwan found that 35.5% of the analyzed bloodstream infections were caused by gram-negative bacteria (GNB). Of those, 18.6% were MDR bacterial strains [22]. Several studies showed a low overall concordance of colonization germs and invasive infectious agents. A retrospective study of neonates with suspected sepsis in Haiti found 20.6% matching bacteria between rectal and blood isolates and similarly to our study a low PPV and high NPV [23]. A systematic review published in 2018 analyzing the results of 27 studies examined the concordance of rectal/skin swabs and bloodstream infections with respect to GNB. They found 7.9% concordant bloodstream infections with GNB in colonized infants. As a consequence, authors were not able to derive recommendations for general GNB screening [10]. Given low concordance of the colonizing pathogens and infectious agents on the one side and the high colonization rates versus the low numbers of infections with MDR bacteria on the other, it is necessary to question the merit of current screening practice for decision making in antibiotic stewardship and the use of last resort antibiotics. Härtel et. al. (2020) found that weekly colonization screening had no impact on sepsis-related mortality, despite decreasing sepsis rates. Instead, the colonization screening lead to a decline in the use of cefotaxime and increasead application rate of meropenem—a third line antibiotic [24]. Schmeh et al. (2019) confirmed the unnecessary use of last resort antibiotics as a major risk for development of antibiotic resistances, since it cannot be determined with certainty which pathogen was causative when multiple germs occur and the frequency of MDR bacterial infections is generally low [16]. In addition to its undisputed role for clinical hygiene, our data suggest that colonization screening may additionally have some relevance for the choice of antibiotic regimens in neonatal pneumonia. At the latest, if the patient deteriorates after 48 h on initial antibiotic therapy, the germ found in the colonization screening should be included in the antibiotic choice.

As most studies have focused on the association between screening and sepsis, available studies for comparison with our pneumonia study arm is limited. Other researchers examined clusters of infections categorized as healthcare-associated infections (HAI) including respiratory infections, blood stream infections, infections of the urinary tract, meningitis, endocarditis, and many more, which leads to less differentiated information.

A recent study from 2020 examined MDR bacteria as infectious agents of ventilator-associated pneumonia (VAP). They found that 39.2% of VAP episodes were caused by MDR pathogens, which is a rate twice as high compared to ours. The higher resistance rate in this Taiwanese study could be explained by a different susceptibility situation in Taiwan compared to Germany. Additionally, the authors discovered that neonates with MDR bacterial VAP indeed had a higher likelihood in receiving insufficient first line antibiotic therapy, while outcomes were equal to non-MDR bacteria-associated VAPs [25]. This suggests that secondary adjusted antibiotic therapy had no disadvantage in VAPs in this population.

In addition to the concordance of colonization and infectious agents, many studies are primarily concerned with the identification of risk factors for the acquisition of MDR bacteria. We found a significant association between the acquisition of MDR bacteria/SERMA and the two variables late-onset pneumonia (LOP, pneumonia later than 7 days after birth) and cesarean section. Newborns with a LOP showed a significantly higher risk of acquiring colonization with MDR bacteria or SERMA. The odds were increased by a factor of 4.8. This finding fits well with the existing literature. Several studies showed a significant correlation with prolonged hospital stay as a result of prolonged exposure to antimicrobial agents [26,27,28]. In addition to being related to the timing of infection, we found that infants born by cesarean section were significantly more likely to acquire colonization with MDR bacteria or SERMA than newborns delivered vaginally. This might be caused by the exposure to different microbiota during birth. Various studies showed possible reasons for the higher resistance of vaginally delivered infants to pathogens. Vaginally delivered infants adopt bacteria from the maternal vaginal microbiome [29] have higher levels of immune mediators such as cytokines [30], and higher microbial diversity of Actinobacteria as well as Bacteroides and lower abundance of Firmicutes. One explanation for this difference might be the higher likelihood of antibiotic use for mothers that undergo cesarean section [31]. However other studies did not show this correlation [20,28,32]. However, Smith et al. (2010) reported an association between vaginal delivery and bloodstream infection with GNB [14]. In contrast to the results of other studies, we did not find any association of the maturity level of the newborns and the risk of acquiring MDR bacterial colonization or infection [3,20,26,27,28,32]. Haase et al. (2014) detected low gestational age and need for mechanical ventilation as risk factors for colonization with MDR bacteria [20], Sakai et al. (2020) found an association with prematurity, extremely low birth weight, and non-exclusive breastfeeding [32]. Auriti et al. (2003) found a higher risk of contracting hospital acquired infections for infants with a very low birth weight, low gestational age and intravascular catheters [3]. Similar to previous studies by others, we could not find a significant difference of colonization rates between sexes [20,27,28].

Our study was based on a representative neonatal cohort that showed similar distribution of infectious agents as published for late-onset sepsis by the German guidelines for bacterial infections in newborns [19]. The frequencies of pneumonia pathogens were also in a comparable range found in similar studies [25,33,34].

Our study has the following limitations: (1) The study is based on retrospective clinical and microbiological data. The data collection was not established with the intention of conducting this study, thus patient information was partly incomlete, and the sample size was small, although the initial evaluation started with over 26,000 microbiological findings. (2) Single-center study results might not be transferable to other institutions. However, the results point to a direction that has implications for evaluating the utility of colonization screening with respect to antibiotic stewardship. (3) The evaluation of transmission dynamics is limited to the weekly screening. It has not been proven if a shorter screening interval would have led to different results. (4) Only blood culture positive sepsis and tracheal aspirate positive pneumonia were considered as markers for infections. This might have led to an underestimation of the true numbers of sepsis and pneumonia. (5) Due to the physical condition of neonates, blood culture volumes are limited (often just one milliliter of blood). This might have led to a higher probability of false negative results. Moreover, the bacterium found might not have been automatically causative for infection. (6) We used deep tracheal aspirate as an accessible and non-invasive technique to identify germs in case of pneumonia. However, the use of TA can lead to overdiagnosis, and a consequently increased use of antibiotics compared to the more invasive bronchioalveolar lavage (BAL) [35,36]. Due to multiple species in one sample, it cannot be determined with certainty which bacterial strain was causative.

## 5. Conclusions

Retrospectively, MDR bacteria and SERMA rarely played a role in neonatal sepsis but were found more frequently as infectious agents of pneumonia. Colonization screening as a predictor of infection (sepsis, pneumonia) with MDR bacteria or SERMA had limited relevance for clinical practice because of the low positive predictive value although the PPV for pneumonia patients of 57.9% is significantly higher than a PPV of 10.0% for sepsis patients. However, due to the high negative predictive value, infections with MDR bacteria/SERMA are highly unlikely to occur in the absence of these colonizers. Late-onset pneumonia and cesarean section are independent risk factors for developing a colonization with MDR bacteria/SERMA, while neonatal colonization surveillance supports the implementation of infection control measures and aids the choice of antibiotic treatment regimens in case of infection in these patients. When the colonization screening is negative, there is no need for last resort antibiotics. On the other hand, clinical misinterpretation of microbiological screening data might rather increase the risk for overtreatment with last resort antibiotics. Therefore, neonatal antibiotic stewardship should consider that in our setting, MDR bacteria rarely cause sepsis and that the colonization with MDR bacteria/SERMA might not be an adequate predictor of pneumonia with these colonizers.

## Figures and Tables

**Figure 1 antibiotics-12-00189-f001:**
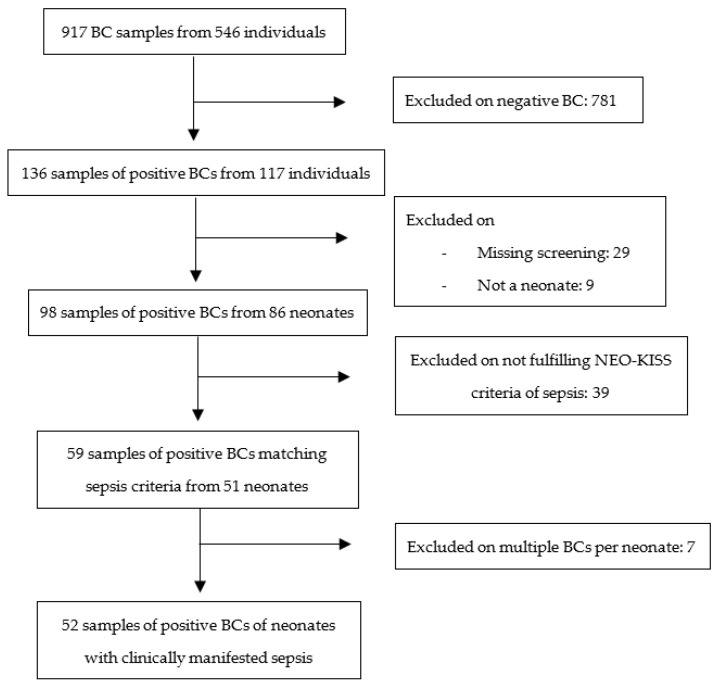
Flowchart and study selection of sepsis patients study arm. Legend: BC = blood culture.

**Figure 2 antibiotics-12-00189-f002:**
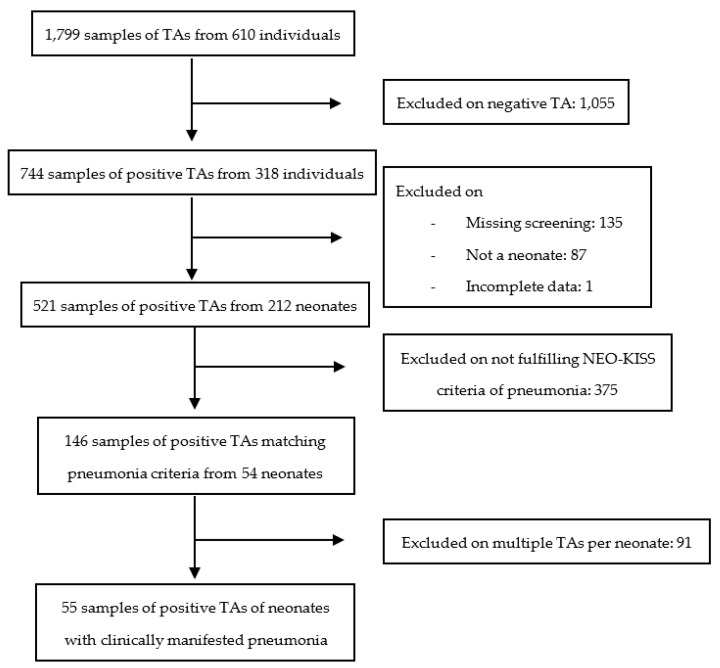
Flowchart and study selection of pneumonia patients study arm. Legend: TA = tracheal aspirate.

**Figure 3 antibiotics-12-00189-f003:**
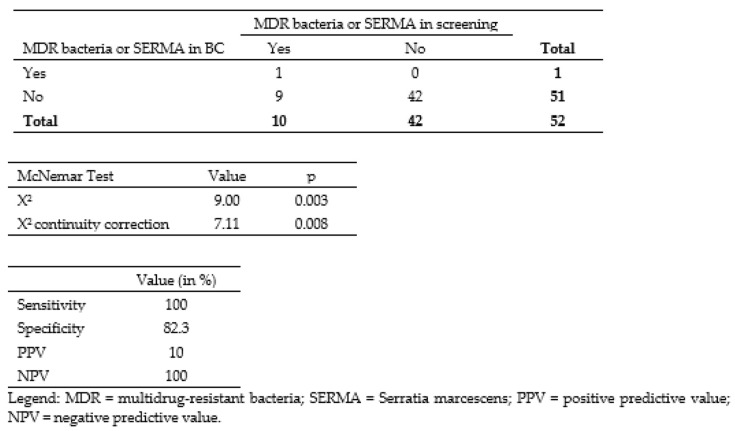
Contingency table with number of patients with MDR bacteria or SERMA in screening and in BC.

**Figure 4 antibiotics-12-00189-f004:**
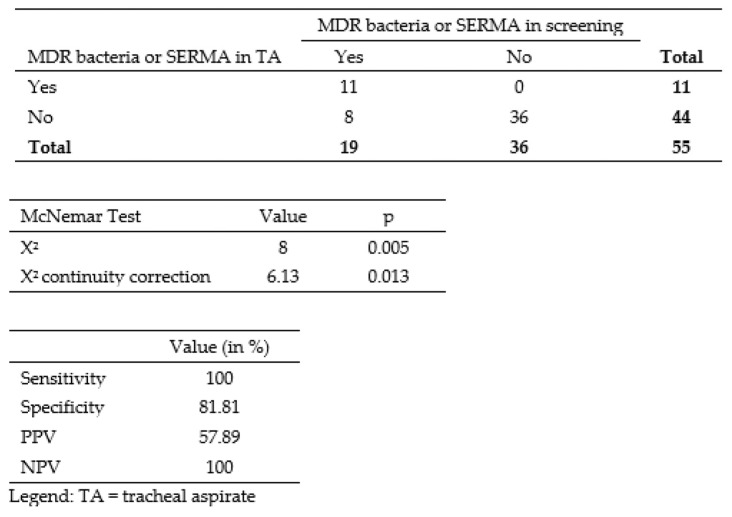
Contingency table with number of patients with MDR bacteria or SERMA in screening and in TA.

**Table 1 antibiotics-12-00189-t001:** Bacteriological results of blood cultures in patients with sepsis and tracheal aspirates in pneumonia patients.

Species	Blood Culture		Tracheal Aspirate	
	*n*	%	*n*	%
**Gram-negative bacteria**				
*Acinetobacter* sp.	0	0	1	1.82
*Citrobacter* sp.	0	0	1	1.82
*Enterobacter cloacae*	1	1.92	3	5.45
*Escherichia coli*	3	5.77	5	9.09
*Klebsiella* spp.	3	5.77	8	14.55
*Pseudomonas* spp.	0	0	3	5.45
*Serratia marcescens*	1	1.92	1	1.82
*Serratia liquefaciens*	0	0	1	1.82
*Stenotrophomonas maltophilia*	0	0	3	5.45
**Gram-positive bacteria**				
Saprophytic germs	0	0	11	20.00
CoNS	34	65.38	14	25.45
*Streptococcus agalactiae*	2	3.85	3	5.45
*Streptococcus pyogenes*	0	0	1	1.82
*Staphylococcus aureus*	3	5.77	3	5.45
*Enterococcus* spp.	4	7.69	8	14.55
*Bacillus cereus*	1	1.92	0	0
**MDR**				
2MRGN NeoPäd	0	0	7	12.73
3MRGN	0	0	2	3.64
VRE	0	0	0	0
MRSA	0	0	1	1.82
**Other**	1	1.92	1	1.82
**Total**	**53**		**77**	

Legend: Multiple species in one sample resulted in more bacterial species than samples of tracheal aspirate/blood cultures. On this account, percentages do not add up to 100 percent. CoNS = coagulase-negative *Staphylococcus*; MDR = multidrug-resistant bacteria; VRE = vancomycin-resistant *Enterococcus*; MRSA = methicillin-resistant *Staphylococcus aureus*; MRGN = multi-resistant Gram-negative bacteria.

**Table 2 antibiotics-12-00189-t002:** Correlation between neonatal risk factors and colonization or infection with MDR bacteria /SERMA in neonates affected by pneumonia.

Patient Group	*n*	MDR Bacteria or SERMA	Χ^2^	*p*	Fisher’s Exact Test	CC	OR	95% CI	
								Lower	Upper
**Gestational age at birth**	55	19	-	-	0.501	0.173	-	-	-
Full-term (≥37 + 0)	18	5							
Middle to late preterm (32 + 0 to 36 + 6)	12	6							
Early preterm (<32 + 0)	25	8							
**Adjusted gestational age**	55	19	-	-	0.296	0.209	-	-	-
Full-term (≥37 + 0)	25	10							
Middle to late preterm (32 + 0 to 36 + 6)	14	6							
Early preterm (< 32 + 0)	16	3							
**Type of pneumonia**	55	19	4.10	0.043 *	-	0.264	4.80	0.955	24.2
EOP ≤ 7 days (ref.)	15	2							
LOP > 7 days	40	17							
**Delivery mode**	52	19	4.90	0.027 *	-	0.293	0.181	0.0357	0.917
Cesarean (ref.)	37	17							
Vaginal	15	2							
**Sex**	55	19	0.02	0.992	-	0.0014	1.01	0.330	3.06
Male	29	10							
Female	26	9							

Legend: CC = contingency coefficient; OR = odds ratio; CI = confidence interval; * Result significant on a 95%-confidence interval.

## Data Availability

The datasets analyzed in this study are available from the corresponding author on request. Data acquisition and interpretation were performed by Alisa Bär in partial fulfillment of the requirements for obtaining the degree “Dr. med.” at the Friedrich-Alexander-Universität Erlangen-Nuernberg, Department of Pediatrics and Adolescent Medicine, Germany.

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
