# Peer review of "Evaluating the Use of Neonatal Colonization Screening for Empiric Antibiotic Therapy of Sepsis and Pneumonia"

_antibiotics, 2023, doi:10.3390/antibiotics12020189_

Round 1

Reviewer 1 Report

Many thanks to the authors for this excellent submission, discussing a topic of very high clinical relevance in the field of neonatal nosocomial infections.

Since 2013, weekly screening for multidrug-resistant gramnegative (MDRGN) bacteria is mandatory in German neonatal intensive care units (NICU). National guidelines recommend considering these colonization analyses for antibiotic treatment regimens.

The authors collected data from all neonatal patients from the introduction of 79 the screening in 2013 until the end of 2021 to answer the following research questions:

The authors analyzed 917 blood and 1,799 tracheal aspirate samples. After applying criteria from the Nosocomial Infection Surveillance for Neonates (NEO-KISS), we included 52 and 55 cases of sepsis and pneumonia, respectively. 19.2% of sepsis patients and 34.5% of pneumonia patients had a prior colonization with MDRGN bacteria or SERMA. In these patients, sepsis was not attributable to MDRGN bacteria yet one SERMA, while in pneumonias ten MDRGN bacteria and one SERMA were identified.

The authors conclude, that colonization screening is a useful tool for hygiene surveillance, but the consideration of colonization with MDRGN/SERMA in case of LOS or LO-pneumonia might promote an inadequate and extensive use of last resort antibiotics in neonates.

In addition, the analysis of risk factors for VAP revealed: Neonates with late-onset pneumonia (LOP) after the first seven days of life showed an increased susceptibility to MDR bacteria/SERMA compared to patients with early-onset pneumonia (EOP) (OR = 4.80, 95% CI 0.955-24.2).

I do have only some minor comments (suggestions) to be considered by the authors in a revised version of this manuscript

(1) Implementation of the KRINKO recommendations

Concerning the KRINKO recommendation from 2013, weekly screening is mandatory in premature infants with a birth weight below 1500 g (VLBW and ELBW) who are treated at a NICU. In this regard, the practice of the authors institution to perform “a weekly screening in every neonatal intensive care patient (irrespective of their gestational age and birth weight) on all three neonatal wards” of this children’s hospital is a local extension of the original KRINKO recommendation. The same applies to the sampling of a  tracheal aspirate from all neonatal patients on ventilation twice weekly

The KRINKO mentions such an extended screening strategy as an adjunctive measure in outbreak managements.

Although maturely born neonates who need intensive care treatment face a risk of nosocomial infection, almost 60% of all NI in this age group are observed in VLBW / ELBW infants. Infections with MDR pathogens in colonized infants after cessation of intensive care treatment are very rare events.

On the other hand, the analysis eventually focuses on infants with LOS or VAP events.

(2) Introduction

“The KRINKO screening follows two objectives: First, the early detection of cross-infections”

Infections are detected by infection surveillance (Neo KISS). The colonisation screening aims at the early detection of nosocomial transmissions

(3) How many patients have been admitted to the NICU during the study period (how many VLBWs) irrespective of blood culture or tracheal aspirate sampling?

(4) Line 231: These contained 744 (41.4%) with positive results from 318 (5.2%) independent individuals. → Please change to 52%

(5) The separate analysis of S. marcescens colonization and infection events is a strength of this study. The KRINKO decided to include S. marcescens into the colonization screening as a pathogen with a higher probability of LOS in colonized patients (e.g. compared to Enterobacter spp.) which has been related to several nosocomial outbreaks and severe cases of LOS with or without meningitis in premature infants.

Two infections (one sepsis and one VAP) were related to S. marcescens.

(6) Line 303/304

The positive predictive value of 58% in tracheal aspirates (concerning the detection of MDR in case of a subsequent VAP) is quite high (compared to 10% concerning the blood culture results). Should there be a comment on that?

In addition (Line 320)

“Besides the 9 cases of 2/3MRGN there was one case of MRSA and one case of SERMA detected. In total, MDR bacterial- or SERMA-associated pneumonia accounted for 20.0% (n = 11) of all pneumonia cases.

This numbers argue against the negative correlation of TA surveillance with TA samples in patients with VAP.

Perhaps the authors should ad a comment on their current calculated treatment strategy in neonates with suspected LO-VAP after this sentence in the discussion:

“…our data suggest that colonization screeening might still have certain relevance for the choice of antibiotic regimens in neonatal pneumonia”

(7) Line 357

Altogether, the colonization rate of or study cohort was 11.5% for MDR bacteria and 7.7% for SERMA.

Does these shares refer to all admissions or to those patients with at least one blood culture or TS screening result?

(8) Final comment concerning ABS

ABS in neonates does not only refer to the decision which antibiotic should be used empirically in patients with LOS or VAP (to the question, whether an individual patient should be treated with meropenem or not). Even if the results of the weekly colonization screening lead to the decision to use meropenem as first line treatment, the antibiotic therapy should be stopped or modified after 48 h in patients with negative blood cultures or the detection of a non MDR pathogen.

Cantey JB, Wozniak PS,  Sanchez PJ. Prospective Surveillance of Antibiotic Use in the Neonatal Intensive Care Unit: Results From the SCOUT Study. Pediatr Infect Dis J 2015; 34: 267-272

In addition, the diagnosis of VAP in NICU patients is very complex and it has been shown that suspected VAP in neonates is a target for ABS. Results from TA should be critically reviewed by the local ABS team (in this study: saprophytic germs, CoNs and Enterococci) in terms of diagnostic stewardship.

Alriyami A, Kiger JR,  Hooven TA. Ventilator-Associated Pneumonia in the Neonatal Intensive Care Unit. NeoReviews 2022; 23: e448-e461

Goerens A, Lehnick D, Büttcher M, Daetwyler K, Fontana M, Genet P, Lurà M, Morgillo D, Pilgrim S, Schwendener-Scholl K, Regamey N, Neuhaus TJ,  Stocker M. Neonatal Ventilator Associated Pneumonia: A Quality Improvement Initiative Focusing on Antimicrobial Stewardship. Frontiers in pediatrics 2018; 6: 262

Iosifidis E, Pitsava G,  Roilides E. Ventilator-associated pneumonia in neonates and children: a systematic analysis of diagnostic methods and prevention. Future Microbiol 2018; 13: 1431-1446

Steinmann KE, Lehnick D, Buettcher M, Schwendener-Scholl K, Daetwyler K, Fontana M, Morgillo D, Ganassi K, O'Neill K, Genet P, Burth S, Savoia P, Terheggen U, Berger C,  Stocker M. Impact of Empowering Leadership on Antimicrobial Stewardship: A Single Center Study in a Neonatal and Pediatric Intensive Care Unit and a Literature Review. Frontiers in pediatrics 2018; 6: 294

(9) I suggest to delete the following section since the study settings and populations are not comparable to the the authors original data.

Dias & Saleem (2019), examined the concordance of colonizers and infectious agents and found that 58% of the neonates developed an infection with the colonizer. 18.2% of those were caused by MDR, bacteria which lead them to the conclusion that MDR screening is of limited utility in forecasting infections [22]. In comparison a study from Taiwan reported a 34.5% incidence of MDR bacteria-related HAIs [23]. This higher incidence of MDR-associated infections might be due to the use of the clustered HAI category on the one hand, and to a different epidemiology between Taiwan and Germany on the other hand.

Author Response

Response to Reviewer 1 Comments

Open Review

English language and style

( ) English very difficult to understand/incomprehensible
( ) Extensive editing of English language and style required
( ) Moderate English changes required
( ) English language and style are fine/minor spell check required
(x) I don't feel qualified to judge about the English language and style

Yes

Can be improved

Must be improved

Not applicable

Does the introduction provide sufficient background and include all relevant references?

(x)

( )

( )

( )

Are all the cited references relevant to the research?

(x)

( )

( )

( )

Is the research design appropriate?

(x)

( )

( )

( )

Are the methods adequately described?

(x)

( )

( )

( )

Are the results clearly presented?

(x)

( )

( )

( )

Are the conclusions supported by the results?

(x)

( )

( )

( )

Comments and Suggestions for Authors

Many thanks to the authors for this excellent submission, discussing a topic of very high clinical relevance in the field of neonatal nosocomial infections.

Since 2013, weekly screening for multidrug-resistant gramnegative (MDRGN) bacteria is mandatory in German neonatal intensive care units (NICU). National guidelines recommend considering these colonization analyses for antibiotic treatment regimens.

The authors collected data from all neonatal patients from the introduction of 79 the screening in 2013 until the end of 2021 to answer the following research questions:

The authors analyzed 917 blood and 1,799 tracheal aspirate samples. After applying criteria from the Nosocomial Infection Surveillance for Neonates (NEO-KISS), we included 52 and 55 cases of sepsis and pneumonia, respectively. 19.2% of sepsis patients and 34.5% of pneumonia patients had a prior colonization with MDRGN bacteria or SERMA. In these patients, sepsis was not attributable to MDRGN bacteria yet one SERMA, while in pneumonias ten MDRGN bacteria and one SERMA were identified.

The authors conclude, that colonization screening is a useful tool for hygiene surveillance, but the consideration of colonization with MDRGN/SERMA in case of LOS or LO-pneumonia might promote an inadequate and extensive use of last resort antibiotics in neonates.

In addition, the analysis of risk factors for VAP revealed: Neonates with late-onset pneumonia (LOP) after the first seven days of life showed an increased susceptibility to MDR bacteria/SERMA compared to patients with early-onset pneumonia (EOP) (OR = 4.80, 95% CI 0.955-24.2).

I do have only some minor comments (suggestions) to be considered by the authors in a revised version of this manuscript

(1) Implementation of the KRINKO recommendations

Concerning the KRINKO recommendation from 2013, weekly screening is mandatory in premature infants with a birth weight below 1500 g (VLBW and ELBW) who are treated at a NICU. In this regard, the practice of the authors institution to perform “a weekly screening in every neonatal intensive care patient (irrespective of their gestational age and birth weight) on all three neonatal wards” of this children’s hospital is a local extension of the original KRINKO recommendation. The same applies to the sampling of a tracheal aspirate from all neonatal patients on ventilation twice weekly

The KRINKO mentions such an extended screening strategy as an adjunctive measure in outbreak managements.

Although maturely born neonates who need intensive care treatment face a risk of nosocomial infection, almost 60% of all NI in this age group are observed in VLBW / ELBW infants. Infections with MDR pathogens in colonized infants after cessation of intensive care treatment are very rare events.

On the other hand, the analysis eventually focuses on infants with LOS or VAP events.

 Response 1:

The KRINKO recommendation from 2013 is an addition to the recommendations from 2007 and 2012 for preterm infants with a birth weight under 1500g. In the section “2.2.1 Who should be screened and how often?” it says: “In contrast to the 2007 recommendation, which explicitly refers to infection prevention in preterm infants with a birth weight of less than 1,500 g treated in intensive care, it may make sense for epidemiological and infection prevention reasons to include all patients treated in a NICU. However, this depends on the specific structures and patient volume of the respective NICU and should be decided locally in consultation with the hospital hygienist and the hygiene commission.”

Since this is indeed not a generally applicable recommendation, as you noted, we have adjusted the relevant text passages (Abstract – Line 24, Introduction – Line 59-62 and Methods – Line 145).

(2) Introduction

“The KRINKO screening follows two objectives: First, the early detection of cross-infections”

Infections are detected by infection surveillance (Neo KISS). The colonisation screening aims at the early detection of nosocomial transmissions

 Response 2: We replaced “cross-infections” with “nosocomial transmissions”.

(3) How many patients have been admitted to the NICU during the study period (how many VLBWs) irrespective of blood culture or tracheal aspirate sampling?

Response 3: In the study period (2013-2021 -> 9 years) we treated 5988 neonates of which 485 were VLBWs. We added this information in the methods (Line 135).

(4) Line 231: These contained 744 (41.4%) with positive results from 318 (5.2%) independent individuals. → Please change to 52%

Response 4: We recalculated all percentages and corrected false values.

 (5) The separate analysis of S. marcescens colonization and infection events is a strength of this study. The KRINKO decided to include S. marcescens into the colonization screening as a pathogen with a higher probability of LOS in colonized patients (e.g. compared to Enterobacter spp.) which has been related to several nosocomial outbreaks and severe cases of LOS with or without meningitis in premature infants.

Two infections (one sepsis and one VAP) were related to S. marcescens

(6) Line 303/304

The positive predictive value of 58% in tracheal aspirates (concerning the detection of MDR in case of a subsequent VAP) is quite high (compared to 10% concerning the blood culture results). Should there be a comment on that?

In Line 370 ff. we added a comment that the PPV for TAs is much higher than for BCs but the screening is still ineffective in the detection of infections with MDR/SERMA due to the significant result in the McNemar Test. The McNemar test shows more significant results the more deviations there are (i.e. screening pos+TA/BC neg or screening neg+TA/BC pos).

In addition (Line 320)

“Besides the 9 cases of 2/3MRGN there was one case of MRSA and one case of SERMA detected. In total, MDR bacterial- or SERMA-associated pneumonia accounted for 20.0% (n = 11) of all pneumonia cases.

This numbers argue against the negative correlation of TA surveillance with TA samples in patients with VAP.

Perhaps the authors should ad a comment on their current calculated treatment strategy in neonates with suspected LO-VAP after this sentence in the discussion:

“…our data suggest that colonization screeening might still have certain relevance for the choice of antibiotic regimens in neonatal pneumonia”

Response 6: You are right, we added a treatment strategy.

(7) Line 357

Altogether, the colonization rate of or study cohort was 11.5% for MDR bacteria and 7.7% for SERMA.

Does these shares refer to all admissions or to those patients with at least one blood culture or TS screening result?

Response 7: We deleted the sentence because it contained the same information as the following sentence. We only analyzed colonization rates for patients with sepsis or pneumonia.

(8) Final comment concerning ABS

ABS in neonates does not only refer to the decision which antibiotic should be used empirically in patients with LOS or VAP (to the question, whether an individual patient should be treated with meropenem or not). Even if the results of the weekly colonization screening lead to the decision to use meropenem as first line treatment, the antibiotic therapy should be stopped or modified after 48 h in patients with negative blood cultures or the detection of a non MDR pathogen.

Cantey JB, Wozniak PS,  Sanchez PJ. Prospective Surveillance of Antibiotic Use in the Neonatal Intensive Care Unit: Results From the SCOUT Study. Pediatr Infect Dis J 2015; 34: 267-272

In addition, the diagnosis of VAP in NICU patients is very complex and it has been shown that suspected VAP in neonates is a target for ABS. Results from TA should be critically reviewed by the local ABS team (in this study: saprophytic germs, CoNs and Enterococci) in terms of diagnostic stewardship.

Alriyami A, Kiger JR,  Hooven TA. Ventilator-Associated Pneumonia in the Neonatal Intensive Care Unit. NeoReviews 2022; 23: e448-e461

Goerens A, Lehnick D, Büttcher M, Daetwyler K, Fontana M, Genet P, Lurà M, Morgillo D, Pilgrim S, Schwendener-Scholl K, Regamey N, Neuhaus TJ,  Stocker M. Neonatal Ventilator Associated Pneumonia: A Quality Improvement Initiative Focusing on Antimicrobial Stewardship. Frontiers in pediatrics 2018; 6: 262

Iosifidis E, Pitsava G,  Roilides E. Ventilator-associated pneumonia in neonates and children: a systematic analysis of diagnostic methods and prevention. Future Microbiol 2018; 13: 1431-1446

Steinmann KE, Lehnick D, Buettcher M, Schwendener-Scholl K, Daetwyler K, Fontana M, Morgillo D, Ganassi K, O'Neill K, Genet P, Burth S, Savoia P, Terheggen U, Berger C,  Stocker M. Impact of Empowering Leadership on Antimicrobial Stewardship: A Single Center Study in a Neonatal and Pediatric Intensive Care Unit and a Literature Review. Frontiers in pediatrics 2018; 6: 294

Response 8: Of course, I agree with you completely. Antibiotic de-escalation after 48 hours or after receipt of microbiology results is immensely important and is usually done this way in our clinic.

(9) I suggest to delete the following section since the study settings and populations are not comparable to the the authors original data.

Dias & Saleem (2019), examined the concordance of colonizers and infectious agents and found that 58% of the neonates developed an infection with the colonizer. 18.2% of those were caused by MDR, bacteria which lead them to the conclusion that MDR screening is of limited utility in forecasting infections [22]. In comparison a study from Taiwan reported a 34.5% incidence of MDR bacteria-related HAIs [23]. This higher incidence of MDR-associated infections might be due to the use of the clustered HAI category on the one hand, and to a different epidemiology between Taiwan and Germany on the other hand.

Response 9: We removed this section due to poor comparability as suggested.

Dear Reviewer,

Thank you very much for the time and effort you have put into our publication. One can see that you have spent a lot of time and effort on our publication. We feel honored, and appreciate your valuable thoughts and feedback.

With kind regards

P. Morhart

Submission Date

25 December 2022

Date of this review

04 Jan 2023 11:17:31

Reviewer 2 Report

Dear Authors,
First of all, let me congratulate you for your novel findings.
I believe that they will be a contribution to the optimization of therapies for complex infectious pathologies.
I would like to make some comments:
Line 54. Correct numbering Reference [4].
Line 61. Please provide references on the increase of nosocomial infection rates in neonates.
Line 69. Please expand "ABS".
Line 125. Please state if any exclusion criteria were used. If not, please indicate.
Line 202. Please check page break
Line 213. In Figure 1 there are several numbers out of memory, please check.
Line 241. I think you should check the diagramming of your figures 1 and 2. 
Congratulations for your results, I am particularly struck by the association with vaginal delivery, you confirm the benefits already studied

Author Response

Response to Reviewer 2 Comments

Open Review

English language and style

( ) English very difficult to understand/incomprehensible
( ) Extensive editing of English language and style required
( ) Moderate English changes required
( ) English language and style are fine/minor spell check required
(x) I don't feel qualified to judge about the English language and style

Yes

Can be improved

Must be improved

Not applicable

Does the introduction provide sufficient background and include all relevant references?

( )

(x)

( )

( )

Are all the cited references relevant to the research?

(x)

( )

( )

( )

Is the research design appropriate?

(x)

( )

( )

( )

Are the methods adequately described?

(x)

( )

( )

( )

Are the results clearly presented?

( )

(x)

( )

( )

Are the conclusions supported by the results?

(x)

( )

( )

( )

Comments and Suggestions for Authors

Dear Authors,
First of all, let me congratulate you for your novel findings.
I believe that they will be a contribution to the optimization of therapies for complex infectious pathologies.
I would like to make some comments:

Point 1:
Line 54. Correct numbering Reference [4].

Response 1: correct numbering done

Point 2:
Line 61. Please provide references on the increase of nosocomial infection rates in neonates.

Response 2: References for increasing nosocomial infection rates in neonates added

Point 3:
Line 69. Please expand "ABS".

Response 3: done

Point 4:
Line 125. Please state if any exclusion criteria were used. If not, please indicate.

Response 4: The sentence “The following inclusion criteria were applied in neonates:” was misleading. The whole paragraph describes the NEO-KISS criteria for sepsis and pneumonia as a basis for the sample selection in 3.1 where the exclusion criteria are mentioned. We removed the misleading phrase.

Point 5:
Line 202. Please check page break

Response 5: Page break removed.

Point 6:
Line 213. In Figure 1 there are several numbers out of memory, please check.

Response 6: We checked the numbers, they should be ok now.

Point 7:
Line 241. I think you should check the diagramming of your figures 1 and 2. 

Response 7: We adjusted the layout of the flowcharts.

Congratulations for your results, I am particularly struck by the association with vaginal delivery, you confirm the benefits already studied

Dear Reviewer,

Thank you very much for the time and effort you have put into our publication. We appreciate your valuable thoughts and feedback.

With kind regards

P. Morhart

Submission Date

25 December 2022

Date of this review

06 Jan 2023 21:59:18